# A novel approach to porcine abnormal sounds recognition based on improved Multi-SVDD

Sunan Zhang[1]*, Bo Jia[2], Yanxin Gao[3]

1 Engineering Training Center, Taiyuan Institute of Technology, Taiyuan, China, 2 Shanxi Animal Husbandry and Veterinary School, Taiyuan, China, 3 Strengthening Foundation Institute, Shanxi Institute of Energy, Taiyuan, China

These authors contributed equally to this work
* zhangsn@tit.edu.cn

## Abstract

During the real-time recognition of porcine abnormal sounds, the accuracy and stability of the recognition method are crucial to guarantee a good performance. For this purpose, an improved Multiple-Support Vector Data Description (Multi-SVDD) is proposed in this paper. Firstly, the improved spectral subtraction using improved Minima Controlled Recursive Averaging (IMCRA) and Spectral Subtraction (SS) is applied to remove the noise of collected sounds. Then, the Mel-Frequency Cepstral Coefficients (MFCC) and first-order differential MFCC (ΔMFCC) are extracted as feature parameters. Finally, the Multi-SVDD is used to detect and recognize the porcine abnormal sounds. In order to improve the accuracy and error-tolerance of Multi-SVDD for human errors on tagging data, the space density information of training data is calculated as the confidences to reduce the interference of outliers in the process of Multi-SVDD training. The experimental results show that the accuracy, precision and recall of the proposed method are as high as 95.0%, 95.4% and 95.0% respectively, which indicates higher error-tolerance capability than classical SVDD.

## 1 Introduction

The sound of animals is the most common bio-signal that can be collected easily from a distance, and will not cause any additional stress to the animals [1]. Therefore, sound analysis has huge potential in interpreting the behavior, health condition, and well-being of animals [2]. Recently, it has been shown that sound recognition played an important role in speech recognition [3], emotion recognition [4], bio-acoustical techniques [5] and among others.

With the increasing use of wireless sensor network technology [6], sound analysis has been widely studied in both wild animals and farm animals. Sound analysis technology used in recognizing the sound of wild animals, such as bird [7], frog [8], anuran [9], has shown good performance. In order to meet the increasing global

**Data availability statement:** The data used in this paper are all from public datasets: https://www.kaggle.com/datasets/titpigrecognition/porcine-scream-sounds-and-cough-sounds/data.

**Funding:** This research work was funded by the Fundamental Research Program of Shanxi Province (Grant No. 202303021222303), Scientific and Technological Innovation Programs of Higher Education Institutions in Shanxi (Grant No. 2023L362) and Taiyuan Institute of Technology Science Research Initial Funding (Grant No. 2022LJ021).

**Competing interests:** The authors have declared that no competing interests exist.

demand for livestock products, livestock management practices have shifted towards the intensive breeding method [10]. Precision livestock farming (PLF) is a trending area in livestock management. Sound analysis has a huge potential for PLF in monitoring the health status of animals [11]. In recent years, sound analysis technology has been used extensively to monitor various kinds of farm animals, like chicken [12], cattle [13], sheep [14] and pig [15] in particular. Many of these research projects have focused on sound analysis of pigs. Chung [2] proposed a pig wasting disease detection and recognition system to detect and classify different kinds of cough sounds due to three types of pig wasting diseases. Cordiero et al. [16] estimated the level of pain in piglets by using a Decision Tree (DT). Xipeng Wang et al. [17] proposed a continuous cough automatic detection method to detect single cough and continuous cough in a complex piggery environment.. A voice activity detection (VAD) method is proposed to automatically segment continuous sound. A multi-classifier fusion strategy is investigated to promote recognition accuracy. Weihao Pan et al. [18] used deep neural network (DNN) and Hidden Markov Model (HMM) theory to recognize pig sound signal. The collected sounds were preprocessed by Kalman filtering and an improved endpoint detection algorithm based on empirical mode decomposition-Teiger energy operator (EMD-TEO) cepstral distance. The 39-dimensional mel-frequency cepstral coefficients (MFCCs) were extracted as characteristic parameters.

Although previous papers have achieved high recognition accuracy, there are still some shortcomings. The acoustic environment in a real pigpen is significantly more complex. In addition to porcine abnormal sounds, there are many other kinds of sounds in the pen. There is a serious imbalance in the number of these kinds of sounds. Consequently, annotating these additional sound types is both time-consuming and challenging, making the accurate detection of porcine abnormal sounds from the entire set of collected sounds particularly difficult. Moreover, recognizing the collected sounds by classification algorithm may classify the other kinds of sounds as porcine abnormal sounds. When training data are incorrectly tagged, the recognition accuracy will be adversely influenced. The recognition method requires high accuracy and stabilization. SVDD is a widely utilized One-Class Classification (OCC) method designed to classify positive cases without well-defined negative cases. SVDD is widely applied in fields such as fault detection [19] and anomaly identification [20]. In this paper, a method based on improved Multi-SVDD is proposed to improve the accuracy and stability during the real-time recognition of porcine abnormal sounds. The valid estimated noise extracted by traditional spectral subtraction may be deficient during the porcine sounds denoising. An improved spectral subtraction using IMCRA and SS is presented to improve the denoising performance in reprocessing. After extracting the MFCC and ΔMFCC as feature parameters, the Multi-SVDD is used to recognize sounds such as porcine cough and scream. In order to improve the error-tolerance of Multi-SVDD for human errors on tagging training data, the space density information of training data are calculated as the confidences to reduce the interference of outliers in the process of Multi-SVDD training. The experimental results show that the accuracy, precision and recall of the proposed method reach 95.0%, 95.4% and 95.0% respectively. When the training

data has tag errors, the proposed method shows higher error-tolerance capability compared to classical SVDD. Meanwhile, the method can be extended to sound recognition of other animal species as well as anomaly detection across various domains, thereby demonstrating broad application potential.

The paper is organized as follows: Section 2 describes the experimental setup and the automatic recognition method based on the improved Multi-SVDD. Section 3 presents the experimental results and comparison between the traditional method and improved method. Section 4 concludes this paper.

## 2 Materials and methods

### 2.1 Materials

In this paper, the experimental data were collected from a large-scale pig farm located in Shanxi Province, China. The sounds were collected through an acoustic pickup device (ELITE model OS-100 made in China). The sampling frequency is 8kHz. For recognizing suspected abnormal pigs, cough and scream of pigs were selected as abnormal sounds. Cough [21] is an early symptom of respiratory diseases, such as asthma and bronchitis. Screams [22] are stress reaction of pigs when they are suddenly hurt. The waveforms and spectrograms of porcine cough and screams are shown in Fig 1 and Fig 2.

In the spectrograms, color is used to represent amplitude. Specifically, brighter colors indicate higher amplitudes, while darker colors correspond to lower amplitudes. It can be seen that there is a difference in the waveforms and spectrograms between cough and screams. Therefore, the cough and screams of pig can be recognized by sound recognition method.

### 2.2 Methods

The proposed method in this paper consists of three stages: preprocessing of porcine sounds, features extraction and abnormal sounds recognition. (1) Sounds preprocessing is composed of activity detection, noise removal, endpoint detection and windowing. (2) The MFCC and ΔMFCC are used for features extraction. (3) An improved Multi-SVDD is proposed to recognize the porcine abnormal sounds in the third stage.

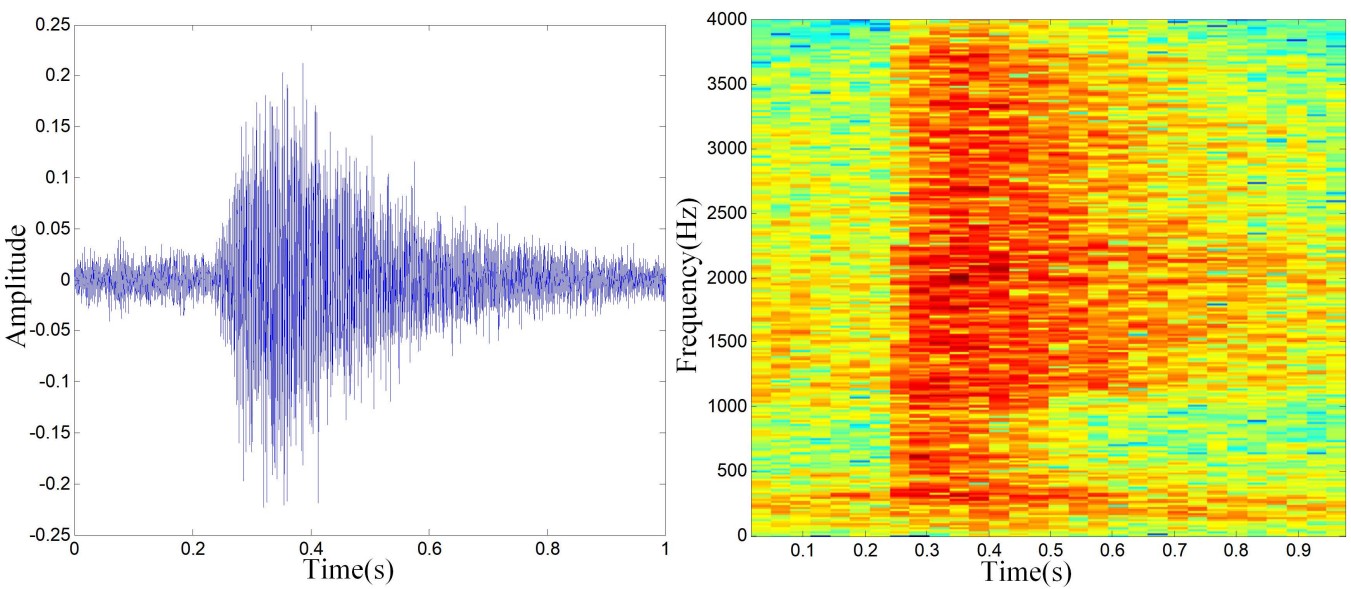

**Fig 1. Waveforms (left) and Spectrograms (Right) of porcine cough.**

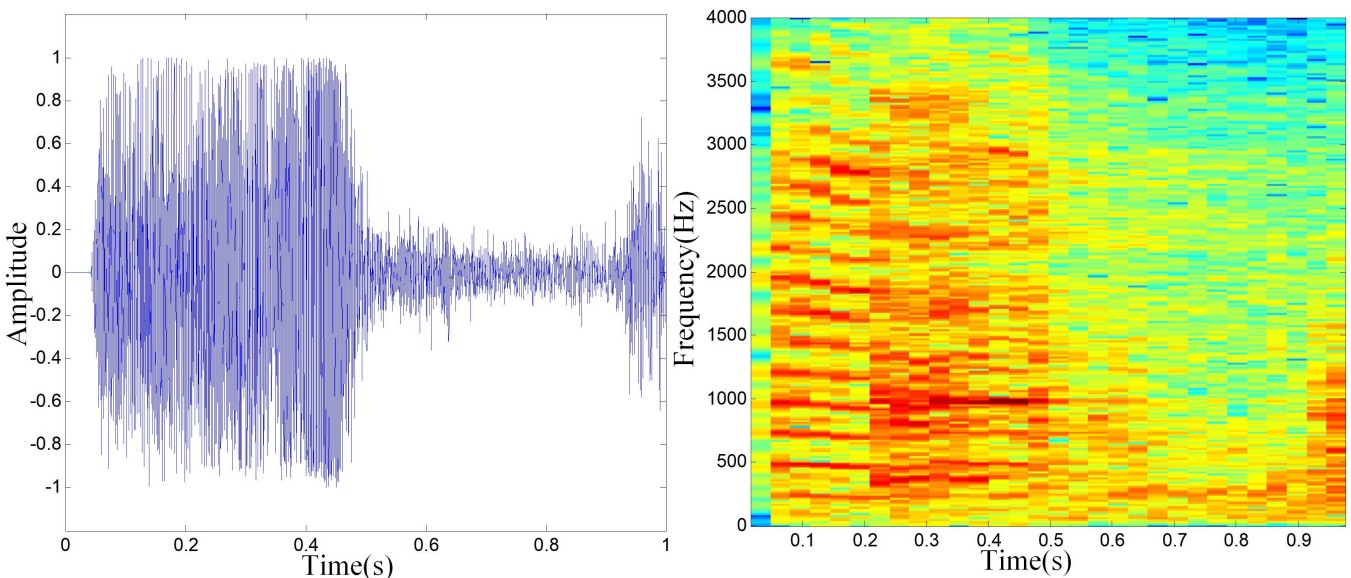

**Fig 2. Waveforms (Left) and Spectrograms (Right) of porcine screams.**

### 2.2.1 Sound preprocessing.

(1)    Activity Detection

The sounds collected in real-time also include noise from the background, which are low in activity level and will affect the accuracy of the sound analysis. In order to eliminate these irrelevant background noise from the collected sounds, the sound energy is calculated, which is defined as [23]:

$$E = \sum_{n=1}^{Len} [x(n)]^2$$

(1)

where $E$ is the sound energy; $x(n)$ is the collected sound and $Len$ is the length of the collected sound.

By setting a threshold value, the background sounds, which has sound energy lower than the threshold, can be excluded from the detection and recognition process. To find an optimum threshold value, the average energies of the two kinds of abnormal sounds and the average energy of ambient noise without porcine sounds are calculated and shown in Fig 3.

It can be seen that the energies of porcine cough and scream with the length of 1s are above 200 and the energy of the sound collected in quiet environment is less than 40. Therefore, the energy threshold is set as 40 in this paper.

(2) Noise Removal

During the process of sound collection and transmission, porcine sounds may be contaminated with noise. The recognition result can be influenced by noise. The background noise in the pigpen is mainly the sound of air blowers. The interference noise is introduced in sound transmission from the pickup to the industrial computer. These noises are all additive noises or stationary sounds. Spectral subtraction is applicable to eliminate the stationary and additive noise [24]. The porcine sound with noise $y(n)$ can be written as:

$$y(n) = x(n) + d(n) \qquad 0 \leq n \leq N-1$$

(2)

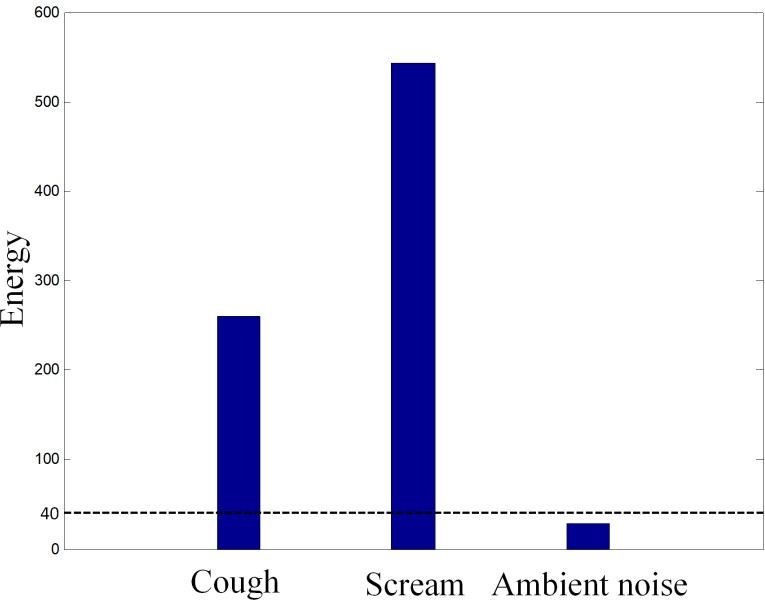

**Fig 3. The energies of different sounds and the energy threshold.**

where $x(n)$ is the porcine sound without noise; $d(n)$ is the noise and $N$ is the length of the sound.

Fast Fourier Transform (FFT) is carried out for the sounds after framing which segment the sound into frames, and the following equations are obtained:

$$Y(l, k) = X(l, k) + D(l, k) \tag{3}$$

where $l$ is the index of frame; $k$ is the index of frequency bin.

The power spectrum of both sides of the equation can be obtained as follows:

$$|Y(l, k)|^2 = |X(l, k)|^2 + |D(l, k)|^2 + X(l, k)D^*(l, k) + X^*(l, k)D(l, k) \tag{4}$$

where $|Y(l, k)|^2$ is the power spectrum of $y(n)$; $|X(l, k)|^2$ is the power spectrum of $x(n)$; $|D(l, k)|^2$ is the power spectrum of $d(n)$; $*$ is the complex conjugate.

Since $x(n)$ and $d(n)$ are independent of each other,

$$E[X(l, k)D^*(l, k)] = E[X^*(l, k)D(l, k)] = 0 \tag{5}$$

The power spectrum can be expressed as follows:

$$|Y(l, k)|^2 = |X(l, k)|^2 + |D(l, k)|^2 \tag{6}$$

Therefore, the denoised sound can be calculated as,

$$|\hat{X}(l, k)|^2 = \begin{cases} |Y(k, l)|^2 - |\hat{D}(k, l)|^2, & |Y(l, k)|^2 \geq |\hat{D}(l, k)|^2 \\ 0, & |Y(l, k)|^2 < |\hat{D}(l, k)|^2 \end{cases} \tag{7}$$

where $\left|\hat{D}(k,l)\right|^2$ is the estimated power spectrum of noise.

During denoising of traditional SS, if the estimated power spectrum of noise is different from the actual noise, 'music noise' may be generated. In order to suppress the 'music noise', the denoised sound can be calculated as follows [25]:

$$\left|\hat{X}(l,k)\right|^2 = \begin{cases} \left|Y(l,k)\right|^2 - \alpha\left|\hat{D}(l,k)\right|,^2 & \left|X(k,l)\right|^2 \geq \alpha\left|\hat{D}(k,l)\right|^2 \\ \beta\left|\hat{D}(l,k)\right|^2, & \left|X(k,l)\right|^2 < \alpha\left|\hat{D}(k,l)\right|^2 \end{cases} \tag{8}$$

where $\alpha$ is reduction factor and $\beta$ is gain compensation factor.

The estimated noise is acquired by extracting 'non-sound frame'. The duration of the collected porcine sound in real time is 1s. Therefore, it is hard to estimate the noise accurately. In this study, Improved Minima Controlled Recursive Averaging (IMCRA) is applied to estimate the noise. The de-noise processing is shown in Fig 4.

IMCRA is a noise estimation method which tracks the noise region by the estimated sound presence probability. The noise is estimated by recursively averaging past spectral power values of the noisy measurement [26]. Under porcine sound presence uncertainty, the recursive averaging $\bar{\lambda}_d(l+1,k)$ is calculated by the conditional sound presence probability, which is shown to be [27]:

$$\bar{\lambda}_d(l+1,k) = \hat{\alpha}_d(l,k)\lambda_d(l,k) + [1-\hat{\alpha}_d(l,k)]\left|Y(l,k)\right|^2 \tag{9}$$

where $\hat{\alpha}_d(l,k)$ is a time-varying frequency-dependent smoothing parameter. It can be obtained by

$$\hat{\alpha}_d(l,k) = \alpha_d + (1-\alpha_d)p(l,k) \tag{10}$$

where $\alpha_d$ is a smoothing parameter; $p(l,k)$ is the conditional sound presence probability. The noise estimation $\hat{\lambda}_d(l+1,k)$ is given by

$$\hat{\lambda}_d(l+1,k) = \varepsilon \cdot \bar{\lambda}(l+1,k) \tag{11}$$

where $\varepsilon$ is a bias compensation factor. In this paper, $\beta = 1.47$

The sound presence probability $p(l,k)$ is estimated by two iterations of smoothing and minimum tracking. The IMCRA shows great performance in estimating noise [28]. Therefore, the power spectrum of noise is estimated by IMCRA to improve the de-noising performance of SS.

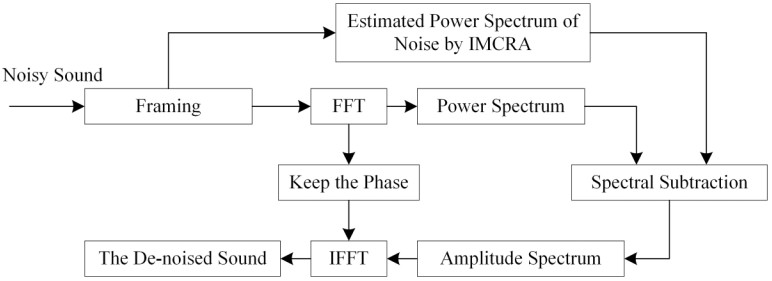

**Fig 4. The processing of improved spectral subtraction.**

(3) Endpoint Detection

In order to find the valid part of the collected sound and reduce the interference of invalid parts, double-threshold end-point detection is used to detect the start point and the end point. The double-threshold can determine the start point and the end point of the collected sounds through short-time average zero-crossing rate and short-time average energy. The short-time average energy is defined as:

$$E_i = \sum_{n=0}^{L-1} y_i^2(n), 1 \le i \le f_n$$

(12)

where $L$ is frame length; $f_n$ is the number of frames; $y_i(n)$ is the $i$th frame of collected sound, which is expressed as:

$$y_i(n) = w(n) * x((i-1) * inc + n)$$

(13)

where $inc$ is the length of frame shift; $w(n)$ is window function, hamming window is selected as window function. The short-time average zero-crossing rate is defined as:

$$Z_i = \frac{1}{2} \sum_{n=0}^{L-1} \left| sgn[y_i(n)] - sgn[y_i(n-1)] \right|, 1 \le i \le f_n$$

(14)

where $inc$ is the length of frame shift; $w(n)$ is window function, hamming window is selected as window function. The short-time average zero-crossing rate is defined as:

$$sgn[y_i(n)] = \begin{cases} 1, y_i(n) \ge 0 \\ -1, y_i(n) < 0 \end{cases}$$

(15)

Taking porcine cough as an example, the starting point and ending point are detected by double-threshold endpoint detection method. The detection result is shown in Fig 5.

In Fig 4, the solid line on the left is the starting point, and the dotted line on the right is the end point of the porcine cough. It can be seen that the starting and ending points of porcine cough can be detected more accurately through the double-threshold endpoint detection method.

(4) Windowing

After endpoint detection, a hamming window is applied to segment the sound into overlapping frames with fixed length [8]. Because the Hamming window has a narrow main lobe and it can reduce the influence of side lobe, the continuity between the porcine sound can be maintained in each frame. The hamming window is defined as:

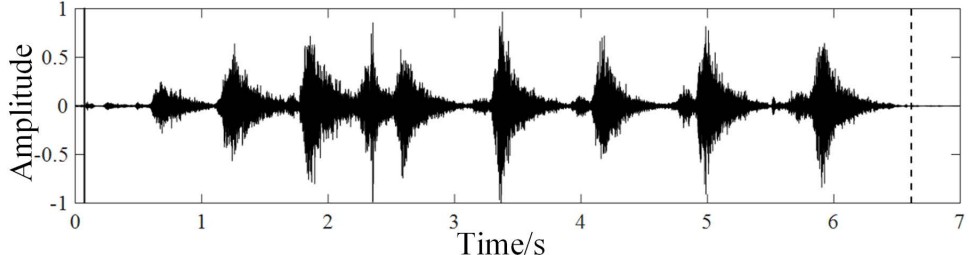

**Fig 5. The endpoint detection of porcine cough.**

$$w(n) = \begin{cases} 0.54 - 0.46 \cos[2\pi n/(L-1)], & 0 \leq n \leq L-1 \\ 0, & \text{otherwise} \end{cases} \tag{16}$$

where $L$ is the length of the frame.

**2.2.2 Feature extraction.** Mel-Frequency Cepstral Coefficients (MFCC) proposed by Davis and Mermelstein, is designed to mimic human auditory response based on the relationship between actual and perceived frequencies [10]. Therefore, it could be a better representation of sound [29]. The relationship between the Mel-frequency and frequency is given as:

$$f_{mel} = 2595\log\left(1 + \frac{f}{700}\right) \tag{17}$$

where $f_{mel}$ is the Mel-frequency of porcine sounds; $f$ is the frequency of porcine sounds.

The extraction steps of MFCC of porcine sound are as follows:

Step1: Preprocess the collected sound, including activity detection, noise removal, end-point detection and windowing.

Step2: FFT is performed on the preprocessed porcine sound, which is expressed as:

$$X(i, k) = \text{FFT}[x_i(m)] \tag{18}$$

where $x_i(m)$ is the $i$th frame of collected sound; $X(i,k)$ is the spectrum of $i$th frame; $k$ is the spectral line number.

The power spectrum $E(i,k)$ is calculated by the spectrum of collected sound, which is defined as:

$$E(i, k) = \left|X(i, k)\right|^2 \tag{19}$$

Step3: The power spectrum $E(i,k)$ of collected sound is filtered through a set of Mel filters, and the energy of the power spectrum of porcine sound signal in the Mel filter bank is obtained as:

$$S(i, m) = \sum_{k=0}^{N-1} E(i, k)H_m(k), 0 \leq m < M \tag{20}$$

where $S(i,m)$ is the energy of Mel filters; $m$ is the serial number of Mel filter, $m = 0,1,\ldots,M\text{-}1$; $M$ is the number of Mel filters; $H_m(k)$ is the transfer function of Mel filter which is shown as:

$$H_m[k] = \begin{cases} 0, & k < f(m-1) \\ \frac{k-f(m-1)}{f(m)-f(m-1)}, & f(m-1) \leq k < f(m) \\ \frac{f(m+1)-k}{f(m+1)-f(m)}, & f(m) \leq k < f(m+1) \\ 0, & k \geq f(m+1) \end{cases} \tag{21}$$

Step 4: Logarithmic operation is performed on the energy of Mel filters $S(i,m)$, as:

$$C(i, m) = \ln[S(i, m)], 0 \leq m < M \tag{22}$$

Step 5: The discrete cosine transform is performed on logarithmic energy of collected sound, which can be described as:

$$mfcc(i, n) = \sum_{m=0}^{M-1} C(i, m) \cos\left[\frac{\pi n(2m+1)}{2M}\right] \tag{23}$$

where *mfcc(i, n)* is the MFCC of collected sound; *i* is the serial number of frames; *n* is the serial number of MFCC. MFCC only reflects the static characteristics of porcine sound, but the first-order difference of MFCC (Δ MFCC) can reflect the dynamic characteristics of the sound. In this paper, 12 dimension MFCC and 12 dimension ΔMFCC are extracted as porcine sound feature parameters.

**2.2.3  The Recognition of Porcine Abnormal Sounds.**  After extracting the feature parameters based on MFCC, the collected sounds are evaluated to see whether they belong to the two kinds of abnormal sounds. Then the collected sounds are classified into two types: cough and scream using Support Vector Data Description (SVDD). SVDD is an one-class classifier. Its basic idea is to construct the smallest sphere which contains all possible training data [30].

In order to detect and classify porcine cough and scream, Multiple-Support Vector Data Description (Multi-SVDD) is proposed in this paper, which is constructed based on two SVDDs. The structure of Multi-SVDD is shown in Fig 6.

In Fig 5, SVDD1 is the recognition model of porcine cough and SVDD2 is the recognition model of porcine scream. $r_1$ is the radius of the first hypersphere; $o_1$ is the center of the first hypersphere; and $r_2$ is the radius of the second hypersphere; $o_2$ is the center of the second hypersphere. Since the hypersphere is constructed based on the training data, it is easily influenced by the tag errors of training data. In order to improve the error-tolerance of Multi-SVDD for human errors on tagging training data, the space density information of training data is calculated as the confidences to reduce the interference of outliers in the process of Multi-SVDD training. The main steps of abnormal sound recognition by improved Multi-SVDD are as follows:

Step 1: The feature parameters of porcine abnormal sounds are defined as $x = \{x_1, \cdots, x_i, \cdots, x_q\}$, where $x_i \in R^d$ ($i = 1, 2, \cdots, q$); $q$ is the number of training data; $d$ is the dimension of feature parameters. In this paper, $d = 24$.

Step 2: Using subtractive clustering [31], the densities of a group of training data $\{x_i\}$ can be defined as:

$$P_i = \sum_{j=1}^{n} \exp\left[\frac{-\|x_i - x_j\|^2}{(0.5r_a)^2}\right], i = 1, 2, ..., q$$

(24)

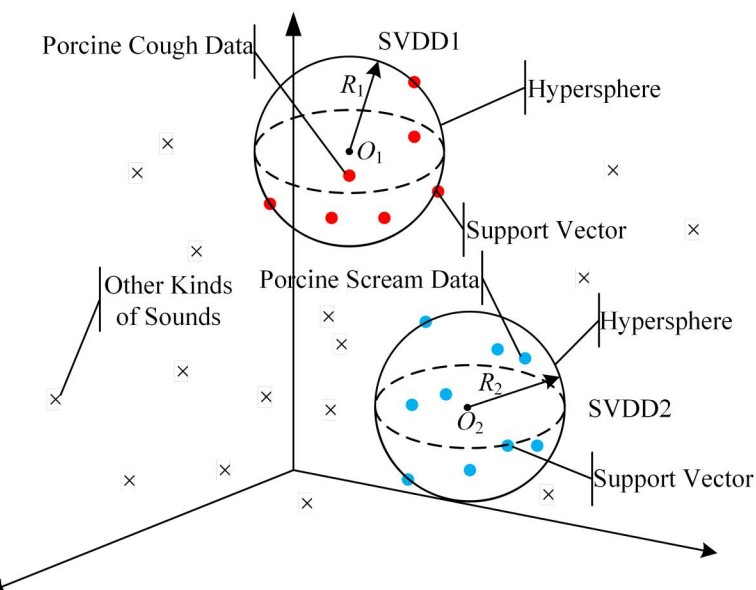

**Fig 6.  The structure of Multi-SVDD.**

where $P_i$ is the density of $x_i$; $r_a$ is the neighboring radius; $q$ is the number of training data. In this paper, $r_a$ is expressed as follows:

$$r_a = \frac{1}{2} \min_j \left\{ \max_i \left\{ \|x_i - x_j\| \right\} \right\}$$

(25)

The density center $P_{max}$ is given by the following:

$$P_{max} = \max(P_i), \ i = 1, 2, \cdots, q$$

(26)

Therefore, the confidences of the training data are defined as:

$$w_i = 1 \left/ \left( 1 + \left( \frac{P_{max} - P_i}{P_{max}} \right)^2 \right) \right., \ i = 1, 2, \cdots, q$$

(27)

Step 3: Let $\phi$ denote the nonlinear mapping function. The process of nonlinear mapping can be expressed as $\phi : R^d \to F^m (d < m)$, where $F^m$ is a high dimensional feature space. The smallest sphere containing all possible training data should be constructed in $F^m$. The confidences of the training data are introduced during the training process of SVDD. The optimization problem can be described as:

$$\min_{r,o,\xi_i} F(r, o, \xi_i) = \min_{r,o,\xi_i} \left( r^2 + C \sum_{i=1}^{q} w_i \xi_i \right)$$

(28)

$$st. \quad \|\phi(x_i) - o\|^2 \leq r^2 + \xi_i \geq 0, \quad \xi_i \geq 0, \ 1 \leq i \leq q$$

(29)

where $r$ is the radius of hypersphere; $o$ is the center of hypersphere; $\xi_i$ is slack variable; $C$ is penalty factor.

Step 4: In order to solve the constraint problem of Eq. (28) and Eq. (29) and introduce Lagrangian multipliers, the formula can be reformulated as:

$$L(r, o, \xi_i, \alpha_i, \beta_i) = r^2 + C \sum_{i=1}^{q} w_i \xi_i - \sum_{i=1}^{q} \alpha_i \left\{ r^2 + \xi_i - \left( \|\phi(x_i)\|^2 - 2o \cdot \phi(x_i) + \|o\|^2 \right) \right\} - \sum_{i=1}^{q} \beta_i \xi_i$$

(30)

where $\alpha_i$ and $\beta_i$ are Lagrange coefficients and $\alpha_i \geq 0$, $\beta_i \geq 0$, $i = 1,2,\ldots,q$.

After solving the equation, the radius can be represented as follows:

$$r^2 = \left( \langle \phi(x_k) \cdot \phi(x_k) \rangle - 2 \sum_{i=1}^{q} \alpha_i \langle \phi(x_i) \cdot \phi(x_k) \rangle + \sum_{i=1}^{q} \sum_{j=1}^{q} \alpha_i \alpha_j \langle \phi(x_i) \cdot \phi(x_j) \rangle \right)$$

(31)

where $x_k$ is support vector.

The kernel function is introduced to replace the inner product, which is defined as $K(x_i, x_j) = \langle \phi(x_i) \cdot \phi(x_j) \rangle$. The formula of radius can then be written as:

$$r^2 = K(x_k, x_k) - 2 \sum_{i=1}^{q} \alpha_i K(x_k, x_i) + \sum_{i=1}^{q} \sum_{j=1}^{q} \alpha_i \alpha_j K(x_i, x_j)$$

(32)

In this paper, radial basis function is selected as the kernel function, which is defined as:

$$K(x_i, x_j) = e^{\frac{-\|x_i - x_j\|^2}{2\sigma^2}}$$

(33)

where $\sigma$ is kernel parameter.

Step 5: When generating a new test data $z$, the decision function for porcine abnormal sounds can be constructed as follows::

$$f(z) = \text{sgn} \left[ r^2 - K(z, z) + 2 \sum_{i=1}^{q} \alpha_i K(z, x_i) - \sum_{i=1}^{q} \sum_{j=1}^{q} \alpha_i \alpha_j K(x_i, x_j) \right]$$

(34)

The decision results of improved Multi-SVDD are the combination of decision functions of SVDD1 and SVDD2. The recognition strategies of improved Multi-SVDD are shown in Table 1.

## 3 Results and discussion

In order to validate the effectiveness of the proposed methods, sounds of 1s duration are used during the course of experiment. After pre-processing, the results of feature extraction and recognition are analyzed in this section. All of the experiments are implemented using Matlab, where we run our algorithm and other experiments on a 8GB NVIDIA GeForce RTX 3070, 2.1GHz Intel Core i7-12700F CPU, and 32GB RAM.

### 3.1 Results of feature extraction

The feature parameters represent the information contained in a sample of porcine sound. Therefore, the same type of porcine sounds have the same feature parameters, while the sounds from different categories are different. After pre-processing, the MFCC and ΔMFCC extracted from porcine cough and porcine scream are shown in Fig 7 and Fig 8 respectively.

The dimensions of MFCC and ΔMFCC are both 12. As shown in Fig 6, the MFCC and ΔMFCC parameters extracted from different frames are basically the same for porcine cough. Similar result is observed for porcine scream as illustrated in Fig 7. However, distinct difference in MFCC and ΔMFCC between cough and screen is observed. Therefore, the MFCC and ΔMFCC are effective to differentiate and represent porcine cough and scream.

The common frequency domain features include MFCC, Linear Prediction Cepstral Coefficient (LPCC), Cochlear Filter Cepstral Coefficients (CFCC) among others. In order to validate the advantages of MFCC and ΔMFCC, recognition accuracy is used to compare the results of different feature parameters. The recognition model is Multi-SVDD and the test data are 200 (100 porcine cough sounds and 100 porcine scream sounds). The recognition results are shown in Table 2.

Table 2 shows that the recognition accuracies of MFCC and ΔMFCC are 91.00% and 94.00%. It achieves the highest recognition accuracies. Therefore, MFCC+ΔMFCC surpasses the other feature parameters for the recognition of porcine abnormal sounds. In this paper, MFCC+ΔMFCC is selected as the feature parameter of porcine abnormal sound.

**Table 1. The decision result of improved Multi-SVDD.**

| The value of decision function | | The decision result |
|---|---|---|
| SVDD1 | | SVDD2 |
| 1 | −1 | Porcine cough |
| −1 | 1 | Porcine scream |
| −1 | −1 | Other kinds of sounds |
| 1 | 1 | Unable to recognize |

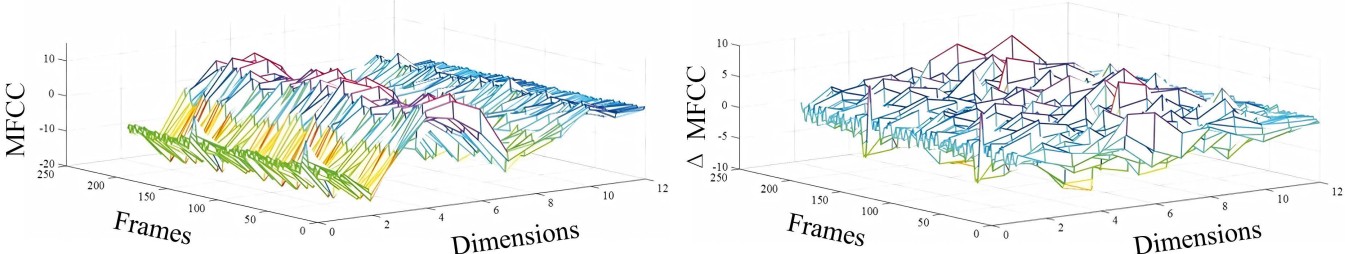

**Fig 7. Feature parameters of porcine cough.**

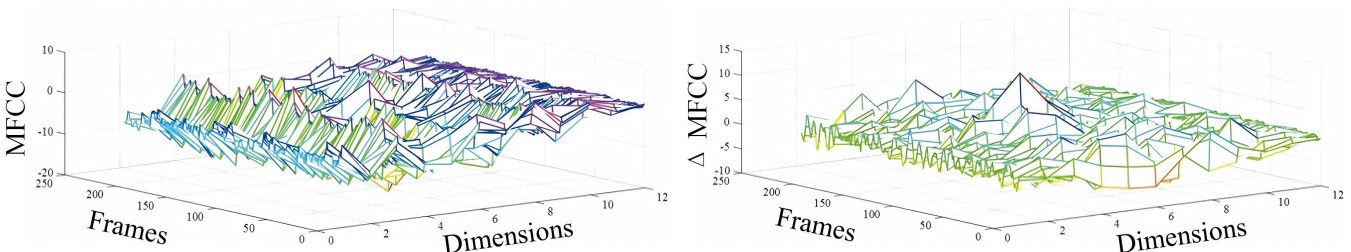

**Fig 8. Feature parameters of porcine scream.**

**Table 2. Comparison of recognition results across different feature parameters.**

| Feature parameters | Recognition accuracy (%) | |
|---|---|---|
| | **Porcine cough sound** | **Porcine scream sound** |
| MFCC | 89.00 | 92.00 |
| MFCC+ΔMFCC | 91.00 | 94.00 |
| LPCC | 78.00 | 81.00 |
| LPCC+ΔLPCC | 81.00 | 84.00 |
| CFCC | 68.00 | 71.00 |
| CFCC+ΔCFCC | 71.00 | 78.00 |
| MFCC+LPCC | 91.00 | 92.00 |
| MFCC+CFCC | 87.00 | 92.00 |
| LPCC+CFCC | 79.00 | 81.00 |

### 3.2 Recognition results of porcine abnormal sounds

In order to evaluate the recognition result quantitatively, accuracy, precision and recall [32] are used as performance measurements.

$$Accuracy = \frac{True\ positive + True\ negative}{Positive + Negative} \times 100\%$$

(35)

$$Precision = \frac{True\ positive}{True\ positive + False\ positive} \times 100\%$$

(36)

$$Recall = \frac{True\ positive}{True\ positive + False\ negative} \times 100\% \tag{37}$$

where *True positive* is the number of target sounds which are correctly recognized; *True negative* is the number of other kinds of sounds which are correctly recognized; *False positive* is the number of other kinds of sounds recognized as target sounds; *False negative* is the number of target sounds recognized as other kinds of sounds.

In order to analyze the recognition performance of improved SVDD when the training data are incorrectly tagged, the contract experiment is set in this study. Taking porcine scream for example, the recognition results of SVDD and improved SVDD with different numbers of tag errors of training data are shown in Table 3. In the contract experiment, 200 porcine scream sounds are selected as training data which are more than 6000 frames after windowing. 100 porcine scream sounds and 200 other kinds of sounds (human voices and other kinds of porcine sounds) are used as test data.

Table 3 shows that the accuracy between SVDD and improved SVDD are the same when the number of tag errors is 0. As the number of tag errors grows, the advantages of improved SVDD are more obvious. The improved SVDD has higher error-tolerance capability than SVDD, which is more likely to misrecognize other kinds of sounds as porcine scream.

In order to intuitively compare the recognition results between SVDD and improved SVDD with different numbers of tag errors of training data, the histograms are shown in Fig 9.

The improved Multi-SVDD consists of two improved SVDD. In order to test the performance of improved Multi-SVDD, the comparison experiment is set in this paper. In the experiment, 200 porcine cough sounds and 200 porcine scream sounds are used as training data, 200 porcine abnormal sounds (100 per kind of porcine abnormal sounds) and 100 other kinds of sounds are used as test data.

Before the training of improved Multi-SVDD, the penalty factor $C$ and kernel parameter $\sigma$ need to be specified. The Particle Swarm Optimization (PSO) is used to optimize these two parameters. In this paper, swarm size is 60, maximum iterations is 200, acceleration coefficients are 1.5 and 2.0, inertia factor is 1.0. The negative average recognition accuracy for recognizing porcine abnormal sounds is used as the fitness function, determined through ten-fold cross-validation. The fitness function of each particle is defined as:

$$fitness = \frac{1}{k}\sum_{m=1}^{k} e_m \tag{38}$$

where *fitness* is fitness function; $k$ is the number of folds in cross-validation, in this paper, $k$ is 10; $e_m$ is the negative average recognition accuracy of the $m$th result, which is defined as follows:

$$e_m = -\frac{N_m}{N} \tag{39}$$

where $N_m$ is the number of correctly classified samples in the $m$th result; $N$ is the total number of training samples.

**Table 3. Comparison of recognition results with different numbers of tag errors in the training data.**

| Proportion of tag errors in training data | SVDD Accuracy (%) | Improved-SVDD Accuracy(%) |
|---|---|---|
| 0 | 98.00 | 98.00 |
| 5% | 86.30 | 98.00 |
| 10% | 73.33 | 94.67 |
| 15% | 69.67 | 89.00 |
| 20% | 68.33 | 80.67 |

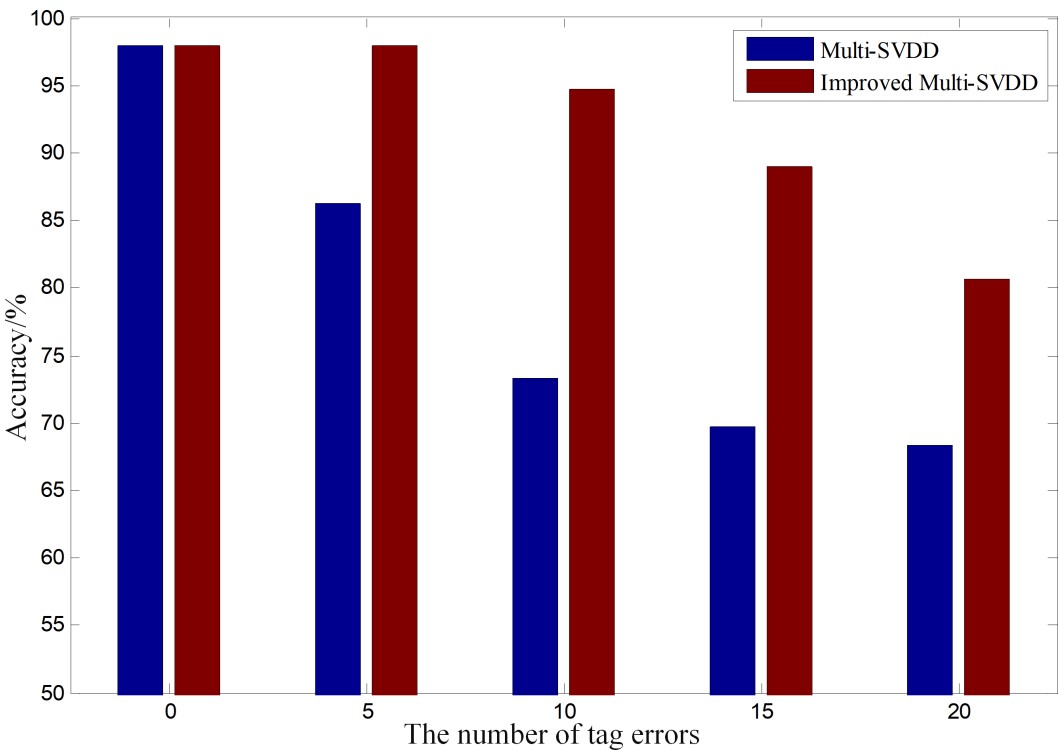

**Fig 9. The recognition results with different numbers of tag errors of training data.**

The optimized penalty factor of SVDD1 is 0.9 and the optimized penalty factor of SVDD2 is 0.25. The optimized kernel parameter of SVDD1 is 1.00 and the optimized kernel parameter of SVDD2 is 7.55. The sound is separated into many frames after pre-processing. The recognition result of the sound is a list which contains the recognition results of all the frames by the improved Multi-SVDD. The majority voting algorithm [33] is employed to find the most elements as the final result. Meanwhile, in order to illustrate the effectiveness of the proposed method, it is compared with the other classification methods [17,23,29]. In the contrast experiment, sounds preprocessing and features extraction of these two methods are the same as this paper. The recognition results are shown in Table 4. The reference feature parameters of porcine abnormal sounds are determined by Fuzzy C-means (FCM) and the test data are classified by comparing with the reference feature parameters in [17]. The abnormal sounds of pigs are detected by SVDD, and then the abnormal sounds are classified as porcine cough and scream by Back-propagation Neural Network (BPNN) in [23]. The classification method of porcine abnormal sounds in [29] is Support Vector Machine (SVM).

It can be seen that the improved Multi-SVDD can recognize porcine abnormal sounds more accurately compared to other methods. The reference feature parameters determined by FCM are difficult to completely characterize the porcine abnormal sounds, which leads to a large detection error. SVDD+BPNN and SVDD+SVM are more likely to misidentify other kinds of sounds as porcine abnormal sounds. The errors of porcine abnormal sounds detection and porcine abnormal sounds classification are superimposed in these two methods. The porcine cough sounds are easy to be recognized as other kinds of sounds by Multi-SVDD. Therefore, the recall of porcine cough sounds is not high enough. These problems can be solved by the improved Multi-SVDD to a certain extent. The accuracy, average precision and average recall are higher than the other three methods. The average recognition time for each sound by the improved Multi-SVDD is 0.0084s, which is lower than that of SVDD+BPNN and SVDD+SVM, and is basically consistent with that of the

**Table 4. Comparison of different classification methods on porcine abnormal sound recognition.**

| Method | Type of sounds | Accuracy(%) | Precision(%) | Recall(%) | Average recognition time(s) |
|---|---|---|---|---|---|
| FCM | Scream | 71.70 | 81.10 | 73.00 | 0.0043 |
| | Cough | | 72.40 | 76.00 | |
| | Other sounds | | 65.70 | 69.00 | |
| | Average | | 73.10 | 72.70 | |
| SVDD+BPNN | Scream | 91.30 | 90.20 | 92.00 | 0.0291 |
| | Cough | | 97.90 | 94.00 | |
| | Other sounds | | 88.00 | 88.00 | |
| | Average | | 92.00 | 91.30 | |
| SVDD+SVM | Scream | 89.30 | 90.10 | 91.00 | 0.0267 |
| | Cough | | 92.70 | 89.00 | |
| | Other sounds | | 85.40 | 88.00 | |
| | Average | | 89.40 | 89.30 | |
| Multi-SVDD | Scream | 93.30 | 98.90 | 94.00 | 0.0142 |
| | Cough | | 98.90 | 88.00 | |
| | Other sounds | | 84.50 | 98.00 | |
| | Average | | 94.10 | 93.30 | |
| Improved Multi-SVDD | Scream | 95.00 | 98.90 | 94.00 | 0.0145 |
| | Cough | | 98.90 | 93.00 | |
| | Other sounds | | 88.30 | 98.00 | |
| | Average | | 95.40 | 95.00 | |

Multi-SVDD. FCM only requires comparison with reference feature parameters for classification, resulting in a relatively short recognition time. The recognition time of the improved Multi-SVDD is capable of meeting the requirements for real-time performance.

## 4 Conclusions

In order to recognize porcine abnormal sounds in real-time accurately and stably, an improved Multi-SVDD is proposed in this paper. Based on the experimental results, the following conclusions can be summarized:

(1) The collected sounds were preprocessed through activity detection, noise removal, endpoint detection and windowing. The valid estimated noise extracted by traditional SS may be deficient during the porcine sounds denoising. IMCRA is applied to estimate the noise. The IMCRA-SS using IMCRA and SS is presented to improve the denoising performance during pre-processing.

(2) After preprocessing, MFCC and first order differential MFCC were extracted as feature parameters. Compared with other feature parameters, MFCC+ΔMFCC are shown to be better for the recognition of abnormal sounds,

(3) In order to recognize porcine cough and scream, Multi-SVDD is proposed in this study. In order to improve the error-tolerance of Multi-SVDD for human errors on tagging training data, the space density information of training data were calculated as the confidences to reduce the interference of outliers in the process of Multi-SVDD training. The proposed improved Multi-SVDD has good performance in terms of accuracy and stability.

In conclusion, the new method proposed in this paper can recognize porcine cough and scream accurately and stably. It demonstrates a highly effective capability in recognizing specific kinds of sounds from the collected unknown sounds. Therefore, this method can also be utilized for the recognition of specific kinds of sounds in other animal species.

Moreover, this method demonstrates application potential in other specific anomaly detection scenarios by extracting the corresponding feature parameters. The drawback of the present paper is that it may cause recognition errors when collecting mixtures of different types of porcine sounds. Additionally, when porcine abnormal sounds are detected, the specific location of the abnormal pigs cannot be determined. Our future work will focus on extracting effective sounds from mishmashed sounds and fusing the abnormal sounds recognition and localization. In future work, the influence of the training sample size on the model's recognition accuracy will be examined, blind source separation methods will be investigated to separate mixed sounds, and the accurate localization of sound sources following porcine abnormal sound recognition will also be explored.

## Author contributions

**Conceptualization:** Sunan Zhang.

**Data curation:** Sunan Zhang, Bo Jia.

**Formal analysis:** Sunan Zhang, Yanxin Gao.

**Investigation:** Sunan Zhang.

**Methodology:** Sunan Zhang.

**Project administration:** Sunan Zhang.

**Resources:** Sunan Zhang, Bo Jia, Yanxin Gao.

**Software:** Sunan Zhang, Yanxin Gao.

**Supervision:** Yanxin Gao.

**Validation:** Sunan Zhang.

**Visualization:** Sunan Zhang, Bo Jia.

**Writing – original draft:** Sunan Zhang.

**Writing – review & editing:** Sunan Zhang.

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
