## [Decision Letter · Decision Letter 0]

24 Jul 2025

PONE-D-25-35613
A novel approach to porcine abnormal sounds recognition based on improved Multi-SVDD
PLOS ONE

Dear Dr. Zhang,

Thank you for submitting your manuscript to PLOS ONE. After careful consideration, we feel that it has merit but does not fully meet PLOS ONE’s publication criteria as it currently stands. Therefore, we invite you to submit a revised version of the manuscript that addresses the points raised during the review process.

We look forward to receiving your revised manuscript.

Kind regards,

Dandan Peng

Academic Editor

PLOS ONE

Journal Requirements:

“This research work was funded by the Fundamental Research Program of Shanxi Province (Grant No. 202303021222303), Scientific and Technological Innovation Programs of Higher Education Institutions in Shanxi (Grant No. 2023L362) and Taiyuan Institute of Technology Science Research Initial Funding (Grant No. 2022LJ021).”

6. We are unable to open your Supporting Information file “Figures.rar”. Please kindly revise as necessary and re-upload.

Reviewers' comments:

Reviewer's Responses to Questions

**Comments to the Author**

1. Is the manuscript technically sound, and do the data support the conclusions?

Reviewer #1: Partly

Reviewer #2: Partly

2. Has the statistical analysis been performed appropriately and rigorously? 

Reviewer #1: N/A

Reviewer #2: N/A

3. Have the authors made all data underlying the findings in their manuscript fully available?

Reviewer #1: Yes

Reviewer #2: No

4. Is the manuscript presented in an intelligible fashion and written in standard English?

Reviewer #1: Yes

Reviewer #2: Yes

5. Review Comments to the Author

Reviewer #1: 1.The current manuscript is primarily driven by performance improvement in sound recognition. It is recommended that the Introduction clearly articulate the key challenges in animal sound recognition, such as difficulties in data annotation, ambiguity in abnormal categories, and complex background noise. The necessity of the proposed method should be justified from a scientific problem-solving perspective.

2.The authors should discuss the potential generalizability of the proposed method to other animal species or even to general unstructured acoustic environments. Emphasizing this extension would enhance the impact and applicability of the work.

3.The mathematical derivation of standard SVDD occupies a significant portion of the manuscript. It is suggested to streamline this part by moving the well-known steps to an appendix or omitting them, and instead highlighting the proposed density-weighted modification and its core equations.

4.Parts of Figures 1 and 6–7 appear blurry and lack detailed annotations. It is recommended to improve the visual clarity and label axes explicitly (e.g., time, frequency, MFCC coefficient index) to enhance interpretability.

5.The titles of the tables can be refined for clarity and conciseness. For instance, the title of Table 4 may be revised to: “Comparison of different classification methods on porcine abnormal sound recognition.”

6.The conclusion section should go beyond summarizing results. The authors are encouraged to discuss the practical feasibility of deploying the proposed method at scale, potential directions for improvement (e.g., joint localization and recognition, real-time implementation), and remaining open challenges (e.g., multi-class detection, data scarcity, and inference efficiency).

7.While this work is focused on porcine sound recognition, the proposed formulation is applicable to broader time-series classification and detection tasks. The authors may consider citing related works such as “Fault detection for point machines: A review, challenges, and perspectives” and “Quantification of abnormal characteristics and flow-patterns identification in pumped storage system” to highlight this potential.

8.The open-source link provided in the manuscript is currently inaccessible, and the dataset cannot be downloaded. The authors are advised to update the link or host the dataset on a stable and publicly accessible repository to ensure reproducibility.

Reviewer #2: The paper presents an improved Multi-SVDD framework for real-time recognition of porcine abnormal sounds (cough and scream). However, there are some concerns should be addressed by the authors.

1. The Introduction only briefly mentions deep neural networks. The authors should explain why their approach outperforms deep-learning alternatives to clarify the research motivation.

2. The public URL for the raw data was inaccessible at the time of review.

3. No cross-validation is reported. Please include k-fold cross-validation or a leave-one-session-out evaluation.

4. Line 54: The manuscript claims "real-time recognition" yet provides no latency or computational-resource metrics.

5. Although PSO is used to tune the penalty factor and kernel parameter, key hyper-parameters such as swarm size, maximum iterations, and inertia weights are missing.

6. The dataset is small. Please add a discussion of its limitations.

6. PLOS authors have the option to publish the peer review history of their article (what does this mean?). If published, this will include your full peer review and any attached files.

Reviewer #1: No

Reviewer #2: No

---

## [Author Response · Author response to Decision Letter 1]

25 Aug 2025

Replies to the first reviewer’s comments:

1.The current manuscript is primarily driven by performance improvement in sound recognition. It is recommended that the Introduction clearly articulate the key challenges in animal sound recognition, such as difficulties in data annotation, ambiguity in abnormal categories, and complex background noise. The necessity of the proposed method should be justified from a scientific problem-solving perspective.

Response: We appreciate the reviewer’s valuable suggestions. The key challenges in animal sound recognition have been incorporated into the introduction, specifically detailed in lines 51–56 on pages 2 and 3.

2.The authors should discuss the potential generalizability of the proposed method to other animal species or even to general unstructured acoustic environments. Emphasizing this extension would enhance the impact and applicability of the work.

Response: Thanks for the reviewer’s kind suggestion. We have discussed the advantages of the proposed method in this paper, as well as the domains in which it can be extended and applied. These details have been added in lines 71-73 on page 3 and lines 410-414 on page 19.

3.The mathematical derivation of standard SVDD occupies a significant portion of the manuscript. It is suggested to streamline this part by moving the well-known steps to an appendix or omitting them, and instead highlighting the proposed density-weighted modification and its core equations.

Response: We sincerely appreciate the reviewer’s thoughtful suggestion. The mathematical derivation of the standard SVDD has been omitted, and the key steps of the improved Multi-SVDD have been emphasized. These revisions are detailed in lines 246–283 on pages 11 to 13.

4.Parts of Figures 1 and 6–7 appear blurry and lack detailed annotations. It is recommended to improve the visual clarity and label axes explicitly (e.g., time, frequency, MFCC coefficient index) to enhance interpretability.

Response: Thanks for the reviewer’s kind suggestion. In the spectrograms of Fig 1, color is used to represent amplitude. Specifically, brighter colors indicate higher amplitudes, while darker colors correspond to lower amplitudes. The detailed annotations have been added in lines 90-91 on page 4. The visual clarity of Fig 6 and Fig 7 has been improved.

5.The titles of the tables can be refined for clarity and conciseness. For instance, the title of Table 4 may be revised to: “Comparison of different classification methods on porcine abnormal sound recognition.”

Response: Thanks for the reviewer’s suggestions. The titles of Table 1 to 4 have been refined to more clearly and accurately reflect the content of the tables.

6.The conclusion section should go beyond summarizing results. The authors are encouraged to discuss the practical feasibility of deploying the proposed method at scale, potential directions for improvement (e.g., joint localization and recognition, real-time implementation), and remaining open challenges (e.g., multi-class detection, data scarcity, and inference efficiency).

Response: Thank you for pointing this out. The drawback of the present paper is that it may cause recognition errors when collecting mixtures of different types of porcine sounds. Additionally, when porcine abnormal sounds are detected, the specific location of the abnormal pigs cannot be determined. Our future work will focus on extracting effective sounds from mishmashed sounds and fusing the abnormal sounds recognition and localization. In future work, blind source separation methods will be investigated to separate mixed sounds, and the accurate localization of sound sources following porcine abnormal sound recognition will also be explored. These details have been added in lines 414-421 on pages 19 to 20.

7.While this work is focused on porcine sound recognition, the proposed formulation is applicable to broader time-series classification and detection tasks. The authors may consider citing related works such as “Fault detection for point machines: A review, challenges, and perspectives” and “Quantification of abnormal characteristics and flow-patterns identification in pumped storage system” to highlight this potential.

Response: Thanks for the reviewer’s kind suggestion. SVDD is a widely utilized One-Class Classification (OCC) method designed to classify positive cases without well-defined negative cases. SVDD is widely applied in fields such as fault detection and anomaly identification. It is presented in the introduction in lines 58-60 on page 3. The references are cited in this section.

8.The open-source link provided in the manuscript is currently inaccessible, and the dataset cannot be downloaded. The authors are advised to update the link or host the dataset on a stable and publicly accessible repository to ensure reproducibility.

Response: Thank you for pointing this out. We have updated the dataset again. It can be downloaded normally now.

Replies to the second reviewer’s comments:

1. The Introduction only briefly mentions deep neural networks. The authors should explain why their approach outperforms deep-learning alternatives to clarify the research motivation.

Response: Thanks for the reviewer’s kind suggestion. The acoustic environment in a real pigpen is significantly more complex. In addition to porcine abnormal sounds, there are many other kinds of sounds in the pen. There is a serious imbalance in the number of these kinds of sounds. Consequently, annotating these additional sound types is both time-consuming and challenging, making the accurate detection of porcine abnormal sounds from the entire set of collected sounds particularly difficult. Moreover, recognizing the collected sounds by classification algorithm may classify the other kinds of sounds as porcine abnormal sounds. When training data are incorrectly tagged, the recognition accuracy will be adversely influenced. These problems all limit the accuracy and stability of classification algorithms, including deep neural network, in identifying abnormal sounds of pigs. The method proposed in this paper can recognize porcine cough and scream from all the collected sounds and improve the accuracy and stability of the recognition process. These statements have been added in line 51-58 on pages 2 to 3.

2. The public URL for the raw data was inaccessible at the time of review.

Response: Thank you for pointing this out. We have updated the dataset again. It can be downloaded normally now.

3. No cross-validation is reported. Please include k-fold cross-validation or a leave-one-session-out evaluation.

Response: Thanks for the reviewer’s kind suggestion. The fitness function of PSO adopts ten-fold cross-validation. The fitness function is defined in lines 355-363 on pages 16 to 17.

4. Line 54: The manuscript claims "real-time recognition" yet provides no latency or computational-resource metrics.

Response: Thank you for pointing this out. We have added an introduction to the algorithm execution environment and a comparison of the average recognition time of different methods, including the method proposed in this paper in lines 292-294 and 377-392 on pages 13 and 17 to 19.

5. Although PSO is used to tune the penalty factor and kernel parameter, key hyper-parameters such as swarm size, maximum iterations, and inertia weights are missing.

Response: Thank you for pointing this out. In the paper, swarm size is 60, maximum iterations is 200, acceleration coefficients are 1.5 and 2.0, inertia factor is 1.0. The negative average recognition accuracy for recognizing porcine abnormal sounds is used as the fitness function, determined through ten-fold cross-validation. These details have been added in lines 353-356 on pages 16 to 17.

6. The dataset is small. Please add a discussion of its limitations.

Response: Thanks for the reviewer’s kind suggestion. SVDD is an one-class classifier. Its basic idea is to construct the smallest sphere which contains all possible training data. Therefore, SVDD does not require a substantial amount of training data. Subsequently, we will analyze the impact of the number of training samples on the model's recognition accuracy. These details have been added in lines 418-419 on page 19.

---

## [Decision Letter · Decision Letter 1]

9 Sep 2025

A novel approach to porcine abnormal sounds recognition based on improved Multi-SVDD

PONE-D-25-35613R1

Dear Dr. Zhang,

We’re pleased to inform you that your manuscript has been judged scientifically suitable for publication and will be formally accepted for publication once it meets all outstanding technical requirements.

Kind regards,

Dandan Peng

Academic Editor

PLOS ONE

Additional Editor Comments (optional):

Reviewer #1:

Reviewer #2:

Reviewers' comments:

Reviewer's Responses to Questions

**Comments to the Author**

1. If the authors have adequately addressed your comments raised in a previous round of review and you feel that this manuscript is now acceptable for publication, you may indicate that here to bypass the “Comments to the Author” section, enter your conflict of interest statement in the “Confidential to Editor” section, and submit your "Accept" recommendation.

Reviewer #1: All comments have been addressed

Reviewer #2: All comments have been addressed

2. Is the manuscript technically sound, and do the data support the conclusions?

Reviewer #1: Yes

Reviewer #2: Yes

3. Has the statistical analysis been performed appropriately and rigorously? 

Reviewer #1: N/A

Reviewer #2: N/A

4. Have the authors made all data underlying the findings in their manuscript fully available?

Reviewer #1: Yes

Reviewer #2: Yes

5. Is the manuscript presented in an intelligible fashion and written in standard English?

Reviewer #1: Yes

Reviewer #2: Yes

6. Review Comments to the Author

Reviewer #1: No other concenrs. It could be accepted.

Reviewer #2: (No Response)

7. PLOS authors have the option to publish the peer review history of their article (what does this mean?). If published, this will include your full peer review and any attached files.

Reviewer #1: No

Reviewer #2: No

---

## [Editor Report · Acceptance letter]

PONE-D-25-35613R1

PLOS ONE

Dear Dr. Zhang,

I'm pleased to inform you that your manuscript has been deemed suitable for publication in PLOS ONE. Congratulations! Your manuscript is now being handed over to our production team.

Kind regards,

on behalf of

Dr. Dandan Peng

Academic Editor

PLOS ONE